# A Model-Free Control Scheme for Attitude Stabilization of Quadrotor Systems

**Jaemin Baek** [1] and **Jinmyung Jung** [2,*]

1   Department of Mechanical Engineering, Gangneung-Wonju National University, Wonju 26403, Korea; jmbaek@gwnu.ac.kr
2   Division of Data Science, College of Information and Communication Technology, The University of Suwon, Hwaseong 18323, Korea
*   Correspondence: jmjung@suwon.ac.kr; Tel.: +82-31-229-8671

**Abstract:** This paper presents an extended time-delayed control (ETDC) scheme and applies it to a quadrotor system. The proposed ETDC scheme uses a one-sample delayed information of the system for canceling out the uncertainties and disturbances in nonlinear quadrotor system, which involves a combination of pole-placement term to deal with the pole assignment. Thus, the proposed one requires no prior knowledge about the quadrotor dynamics, which is called model-free control scheme, and then assures fast convergence rate while providing simplicity structure. To suppress time-delayed estimation (TDE) errors generated by using one-sample delayed information of the system, a new auxiliary control scheme is designed in the proposed ETDC scheme. It results in a proper switching gain without undesirable side effect, including chattering and input fluctuation. Moreover, given that it does not require any number of additional parameters, the number of the parameters in the proposed ETDC scheme has no change compared to that in conventional time-delayed control. From these benefits, the proposed one can be recognized as a simple and effective alternative to the quadrotor system with nonlinearity and complexity. The tracking errors are proved to be uniformly ultimately bounded through Lyapunov function. The effectiveness of the proposed ETDC scheme is verified by the simulation with the quadrotor system, which is compared to that of the conventional time-delayed control scheme.

**Keywords:** model-free control; attitude control; quadrotor; unmanned aerial vehicle

## 1. Introduction

The research for the quadrotor systems have been tremendously increased in recent years, which have shown many tasks including rescue [1], surveillance [2], inspection [3], and topography mapping [4]. These tasks require the robust trajectory-tracking because they should maintain the high precision performance. However, given that the quadrotor systems have typical nonlinear systems with coupling dynamics while being light-weight, it can be difficult to achieve the desired tracking performance because of sensitivity. For the satisfactory control performance of the quadrotor systems, several researchers have been fascinated by many control schemes for many years.

Traditionally, the linear control schemes have been widely employed for controlling the quadrotor systems. As one of them, the linear quadratic regulator (LQR) control scheme [5] has been applied to obtain desired tracking performance in the quadrotor systems. The LQR control scheme has required two variables that have a significant effect on the reliability of the controller. However, it is not easy to obtain that an optimal value of these variables at which best reductions in responses is obtained. It means that it may adversely affect robust performance. As another control scheme, $H_\infty$ control scheme [6] has been introduced to remedy aforementioned problem, which enhances the robust performance. This control

scheme is adopted to achieve the optimal gain through Riccati equation or linear matrix inequality, which aims at providing a sense of stability for the hovering of the quadrotor systems while reducing the external disturbances. However, given that it requires the complex procedures, e.g., Riccati equation or linear matrix inequality, in obtaining the optimal gain, this control scheme has difficulty guaranteeing the globally optimal solution in real systems. Furthermore, the above-mentioned control schemes have a potential undesirable side effect when generating the abrupt and unsuspected disturbances and finally may lead to instability in the quadrotor systems.

To improve these problems, conventional sliding-mode control (SMC) schemes [7,8] have been studied in the quadrotor systems, which has been well-established as one of the most important tools in industrial systems, including a robot manipulator [9], electric power steering system [10], and cable-driven manipulator [11]. The outstanding advantage of the conventional SMC scheme improves the robust performance in the presence of uncertain conditions [12], even though the unknown system dynamics exists in the systems. Additionally, these control schemes can be easily applicable to the quadrotor systems owing to the simplicity of their structure. However, given that the conventional SMC schemes employ the time-invariant switching gains required to be greater than the upper-bound on the uncertainties and disturbances, they are very conservative because the information of the upper-bound cannot be perfectly achieved in the real quadrotor systems. In other words, it implies that they may cause undesirable side effects [13], including chattering and input fluctuation, and hence there is a limit to improving the convergence rate without negative results.

To remedy these undesirable side effects, studies on the SMC schemes have been conducted for many years. As one of them, adaptive sliding-mode control (ASMC) schemes [14–17] have been recently developed, which produce appropriate control inputs in adapting to an uncertain dynamics and unknown perturbations efficiently owing to the time-varying switching gains. For this reason, the ASMC schemes aim at overcoming some problems mentioned above. However, given that the time-varying switching gains are only regulated in relation to tracking errors, there is room for them greater than the upper-bound on the quadrotor systems. In other words, although the high switching gains in the ASMC schemes may have positive results generally for the nonlinear systems with large moment of inertia (MOI), including robot manipulators [18,19] and space vehicles [20], it may have a negative impact, e.g., system instability, on the systems with small MOI such as the quadrotor systems because they can be easily shaken by external pressure such as abrupt wind. It implies that it may make a result sensitive to external disturbances. Furthermore, since these ASMC schemes employ equivalent terms that seem to be similar to feed-forward terms, they have no choice but to require exact information of the quadrotor system model, including parametric uncertainties, MOI, and unmodeled disturbances. In other words, given that these control schemes are mostly based on mathematical models that incorporate a priori knowledge of systems, they have difficulty in obtaining an exact and simple model for control design in real quadrotor systems and hence may cause degraded tracking-trajectory control performance while decreasing the robustness. In addition, the number of the parameters in these control schemes should be increased for adjusting the time-varying switching gains appropriately. However, their increase provides difficulties for practicing engineers to use the quadrotor systems. This is a very important factor while designing the controller in the practical aspect. In this regard, it would be meaningful to develop a simple, practical, and powerful control scheme for an effective solution with a fast convergence rate while guaranteeing the system stability in the quadrotor systems.

In this paper, we propose an extended time-delayed control (ETDC) scheme and apply it to the quadrotor system. The proposed ETDC scheme uses the state information from a previous time to cancel out the uncertain and unknown nonlinear dynamics. Therefore, the proposed one requires no prior knowledge about the quadrotor dynamics. Moreover, it is possible to achieve the dominant pole using the pole-placement term so that it can be easily performed in the desired convergence rate. This is a reason that the proposed ETDC scheme assures a fast convergence rate while providing a simple structure. To suppress the time-delayed estimation (TDE) errors generated by using the

state information at a previous time, the proposed one involves the switching gain based on the TDE error at a previous time, as an auxiliary control scheme. Therefore, it provides a proper time-varying switching gain without undesirable side effects, including chattering and input fluctuation. In addition, the proposed ETDC scheme does not require any number of additional parameters while providing these advantages, unlike the existing TDC-based control schemes. From these reasons, the proposed ETDC scheme can be recognized as a simple and effective alternative to the quadrotor system with nonlinearity and complexity. It may be convenient for the practicing engineers who do not have control engineering knowledge. The effectiveness of the proposed ETDC scheme is verified through quadrotor simulations, which is compared to that of existing control schemes.

The remainder of this paper is organized as follows: In Section 2, we briefly introduce the quadrotor system dynamics. In Section 3, we explain how to design both the conventional TDC scheme and the proposed ETDC scheme. In Section 4, we carried out simulations with the quadrotor system. In Section 5, we discuss a future perspective and a supplementary simulation. In Section 6, we conclude with a brief summary of this paper.

## 2. System Dynamics

The quadrotor system dynamics with angular components can be illustrated in accordance with a Euler–Lagrange equation [21–23]. To begin with, the Euler–Lagrange equation can be represented as follows:

$$\frac{d}{dt}\left(\frac{\partial \mathcal{L}}{\partial \dot{\boldsymbol{\eta}}}\right) - \frac{\partial \mathcal{L}}{\partial \boldsymbol{\eta}} = \boldsymbol{\tau}_{\eta}^{T} \tag{1}$$

where $\boldsymbol{\eta} = [\phi, \theta, \psi]^{T}$ describes the roll, pitch, and yaw angles, respectively. $\dot{\boldsymbol{\eta}} = [\dot{\phi}, \dot{\theta}, \dot{\psi}]^{T}$ is defined as the derivative of $\boldsymbol{\eta}$. $\boldsymbol{\eta}$ and $\dot{\boldsymbol{\eta}}$ are set in inertial frame, as shown in Figure 1. $\boldsymbol{\tau}_{\eta} = [\tau_{\phi}, \tau_{\theta}, \tau_{\gamma}]^{T}$ is the roll, pitch, and yaw control torques, respectively. The Lagrangian $\mathcal{L}$ is directly related to angular components and hence can be described as

$$\mathcal{L} = \mathbf{E}_{R} = \frac{1}{2}\boldsymbol{\nu}^{T}\mathbf{I}_{m}\boldsymbol{\nu} = \frac{1}{2}\dot{\boldsymbol{\eta}}^{T}\mathbf{J}_{c}\dot{\boldsymbol{\eta}} \tag{2}$$

where $\mathbf{E}_{R}$ is the rotational kinetic energy and $\boldsymbol{\nu}$ is angular velocity in body frame, as shown in Figure 1. $\mathbf{J}_{c}$ is Jacobian matrix that will be introduced later on. $\mathbf{I}_{m}$ is moment of inertia (MOI) as $\mathbf{I}_{m} = \text{diag}(J_{xx}, J_{yy}, J_{zz})$, where $J_{xx}$, $J_{yy}$, and $J_{zz}$ are the MOI of the $x$-axis, $y$-axis, and $z$-axis in the quadrotor system, respectively. It implies that the quadrotor system is assumed to be a symmetric structure with the four arms aligned with the body axes, i.e., $x$-axis and $y$-axis. The transformation matrix $\mathbf{W}_{\eta}$ for angular velocity from the inertial frame to the body frame can be represented as follows:

$$\mathbf{W}_{\eta} = \begin{bmatrix} 1 & 0 & -s_{\theta} \\ 0 & c_{\phi} & c_{\theta}s_{\phi} \\ 0 & -s_{\phi} & c_{\theta}c_{\phi} \end{bmatrix} \tag{3}$$

where $c_{\phi} = \cos(\phi)$, $s_{\phi} = \sin(\phi)$, $c_{\theta} = \cos(\theta)$, and $s_{\theta} = \sin(\theta)$.

From Equations (1) and (2), we consider the angular dynamic model [21–23] of the quadrotor as follows:

$$\mathbf{J}_{c}\ddot{\boldsymbol{\eta}} + \mathbf{C}(\boldsymbol{\eta}, \dot{\boldsymbol{\eta}})\dot{\boldsymbol{\eta}} = \boldsymbol{\tau}_{\eta} + \boldsymbol{\tau}_{d} \tag{4}$$

where $\ddot{\boldsymbol{\eta}} = [\ddot{\phi}, \ddot{\theta}, \ddot{\psi}]^T$ is the angular acceleration. $\boldsymbol{\tau}_d$ is called the external disturbance (ED) that is assumed to be bounded in this paper, i.e., $\|\boldsymbol{\tau}_d\| \leq \tau_d^*$, where $\tau_d^*$ is a positive constant value. In Equation (2), the Jacobian matrix $\mathbf{J}_c$ can be defined as follows:

$$\mathbf{J}_c = \mathbf{W}_\eta \mathbf{I}_m \mathbf{W}_\eta = \begin{bmatrix} J_{11} & J_{12} & J_{13} \\ J_{21} & J_{22} & J_{23} \\ J_{31} & J_{32} & J_{33} \end{bmatrix} \tag{5}$$

where

$$
\begin{aligned}
J_{11} &= J_{xx} \\
J_{12} &= J_{21} = 0 \\
J_{13} &= J_{31} = -J_{xx}s_\theta \\
J_{22} &= J_{yy}c_\phi^2 + J_{zz}s_\phi^2 \\
J_{23} &= J_{32} = (J_{yy} - J_{zz})c_\phi s_\phi c_\theta \\
J_{33} &= J_{xx}s_\theta^2 + J_{yy}s_\phi^2 c_\theta^2 + J_{zz}c_\phi^2 c_\theta^2,
\end{aligned}
$$

where $J_{xx}$, $J_{yy}$, and $J_{zz}$ are the positive constant values. In Equation (4), the second term of left-hand side follows that

$$\mathbf{C}(\boldsymbol{\eta}, \dot{\boldsymbol{\eta}}) = \begin{bmatrix} C_{11} & C_{12} & C_{13} \\ C_{21} & C_{22} & C_{23} \\ C_{31} & C_{32} & C_{33} \end{bmatrix} \tag{6}$$

where

$$
\begin{aligned}
C_{11} &= 0 \\
C_{12} &= (J_{yy} - J_{zz})(\dot{\theta}c_\theta s_\phi + \dot{\psi}s_\phi^2 c_\theta) + (J_{zz} - J_{yy})\dot{\psi}c_\phi^2 c_\theta - J_{xx}\dot{\psi}c_\theta \\
C_{13} &= (J_{zz} - J_{yy})\dot{\psi}c_\phi s_\phi c_\theta^2 \\
C_{21} &= (J_{zz} - J_{yy})(\dot{\theta}c_\phi s_\phi + \dot{\psi}s_\phi^2 c_\theta) + (J_{yy} - J_{zz})\dot{\psi}c_\phi^2 c_\theta + J_{xx}\dot{\psi}c_\theta \\
C_{22} &= (J_{zz} - J_{yy})\dot{\phi}c_\phi s_\phi \\
C_{23} &= -J_{xx}\dot{\psi}s_\theta c_\theta + J_{yy}\dot{\psi}s_\phi^2 c_\theta s_\theta + J_{zz}\dot{\psi}c_\phi^2 s_\theta c_\theta \\
C_{31} &= (J_{yy} - J_{zz})\dot{\psi}c_\theta^2 s_\phi c_\phi - J_{xx}\dot{\theta}c_\theta \\
C_{32} &= (J_{zz} - J_{yy})(\dot{\theta}c_\phi s_\phi s_\theta + \dot{\phi}s_\phi^2 c_\theta) + (J_{yy} - J_{zz})\dot{\phi}c_\phi^2 c_\theta + J_{xx}\dot{\psi}s_\theta c_\theta - J_{yy}\dot{\psi}s_\phi^2 s_\theta c_\theta - J_{zz}\dot{\psi}c_\phi^2 s_\theta c_\theta \\
C_{33} &= (J_{yy} - J_{zz})\dot{\phi}c_\phi s_\phi c_\theta^2 - J_{yy}\dot{\theta}s_\phi^2 c_\theta s_\theta - J_{zz}\dot{\theta}c_\phi^2 c_\theta s_\theta + J_{xx}\dot{\theta}c_\theta s_\theta
\end{aligned}
$$

where $\mathbf{C}(\boldsymbol{\eta}, \dot{\boldsymbol{\eta}})$ is the Coriolis matrix, containing the gyroscopic and centripetal effects. $\boldsymbol{\tau}_\eta$ mentioned in Equation (4) is directly related to the rotors of the quadrotor system, which can be represented as follows:

$$\boldsymbol{\tau}_\eta = \begin{bmatrix} 0 & l & 0 & -l \\ -l & 0 & l & 0 \\ -\frac{c_d}{b} & \frac{c_d}{b} & -\frac{c_d}{b} & \frac{c_d}{b} \end{bmatrix} \cdot \mathbf{f} \tag{7}$$

where $\mathbf{f} = [f_1, f_2, f_3, f_4]^T$ denotes the forces generated by four rotors of the quadrotor system (Figure 1). $f_i = b\omega_i^2$ where $\omega_i$ is the rate of $i$-th rotor, e.g., $i = 1, 2, 3, 4$, of the quadrotor system. $b$ and $l$ are the lift coefficient and the length from the center of mass to the center of each rotor, respectively. $\frac{c_d}{b}$ is considered as the drag coefficient.

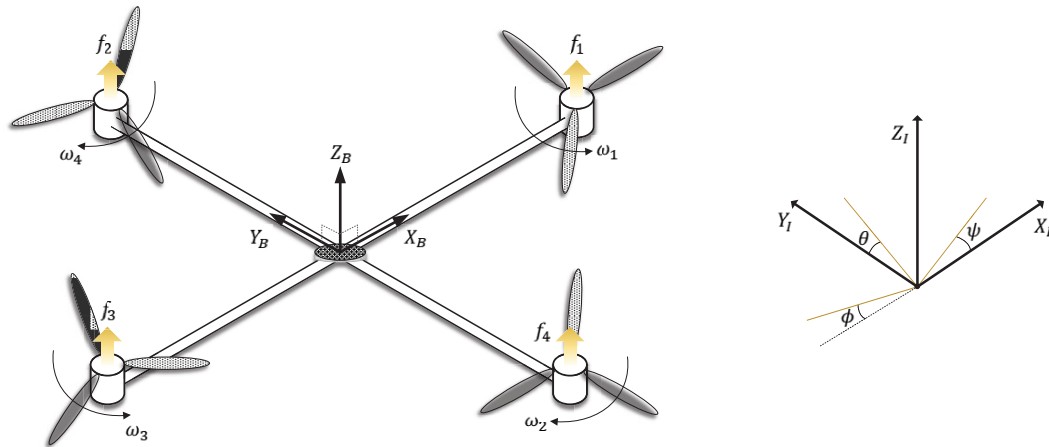

**Figure 1.** A schematic diagram of quadrotor system in the inertia ($\bullet_I$) and body ($\bullet_B$) frames.

## 3. Control Strategies

### 3.1. Conventional Time-Delayed Control Scheme

To achieve a stable attitude control for the quadrotor system, we consider the nonlinear quadrotor system model (Equation (4)). Representing Equation (8) in a compact and simple form yields

$$\ddot{\boldsymbol{\eta}} = \mathbf{N} + \bar{\mathbf{J}}^{-1}\boldsymbol{\tau}_\eta \tag{8}$$

where $\mathbf{N} = \bar{\mathbf{J}}^{-1}[-(\mathbf{J}_c - \bar{\mathbf{J}})\ddot{\boldsymbol{\eta}} - \mathbf{C}(\boldsymbol{\eta},\dot{\boldsymbol{\eta}})\dot{\boldsymbol{\eta}} + \boldsymbol{\tau}_d]$. $\bar{\mathbf{J}} = \text{diag}(\bar{J}_\phi, \bar{J}_\theta, \bar{J}_\psi) = \text{diag}(\bar{J}_1, \bar{J}_2, \bar{J}_3)$ is the TDC gain that will be determined for guaranteeing the system stability later on. $\bar{\mathbf{J}}^{-1}$ is defined as the inverse matrix of $\bar{\mathbf{J}}$. In Equation (8), it is assumed that Euler angles $\boldsymbol{\eta}$ constraints of the quadrotor system model are bounded as follows:

$$-\frac{\pi}{2} < \phi < \frac{\pi}{2}$$
$$-\frac{\pi}{2} < \theta < \frac{\pi}{2} \tag{9}$$
$$-\pi < \psi < \pi.$$

Since $\mathbf{N}$ in Equation (8) is not available, we can use its estimate $\hat{\mathbf{N}}$ as below:

$$\hat{\mathbf{N}} \cong \mathbf{N}_- = \ddot{\boldsymbol{\eta}}_- - \bar{\mathbf{J}}^{-1}\boldsymbol{\tau}_{\eta,-} \tag{10}$$

where $\hat{\mathbf{N}}$ is one-sample delayed information of $\mathbf{N}$, which is called time-delayed estimation (TDE). Subscript "$-$" means the one-sample delayed information of any variables in present time, i.e., $\bullet_{t-L}$. $L$ is set as the sampling period in the quadrotor system, which should be required to be small sufficiently. Then, the one-sample delayed acceleration can be calculated by the simple procedure as follows:

$$\ddot{\boldsymbol{\eta}} = \frac{(\boldsymbol{\eta} - 2\boldsymbol{\eta}_- + \boldsymbol{\eta}_{--})}{L^2} \tag{11}$$

where $\boldsymbol{\eta}_{--}$ is the one-sample delayed information of $\boldsymbol{\eta}_-$.

To obtain attitude stabilization of the quadrotor system, time-delayed control (TDC) scheme [19,24,25] can be expressed while employing Equation (10) as in the following form:

$$\boldsymbol{\tau}_\eta = -\bar{\mathbf{J}}\ddot{\boldsymbol{\eta}}_- + \bar{\boldsymbol{\tau}}_{\eta,-} + \bar{\mathbf{J}}(\ddot{\boldsymbol{\eta}}_d + \mathbf{K}_a \dot{\mathbf{e}} + \mathbf{K}_b \mathbf{e}) \tag{12}$$

where $\boldsymbol{\tau}_\eta$ is called the control torque of TDC scheme. To achieve the dominant pole, $\mathbf{K}_a = \mathrm{diag}(K_{a,\phi}, K_{a,\theta}, K_{a,\psi}) = \mathrm{diag}(K_{a,1}, K_{a,2}, K_{a,3})$ and $\mathbf{K}_b = \mathrm{diag}(K_{b,\phi}, K_{b,\theta}, K_{b,\psi}) = \mathrm{diag}(K_{b,1}, K_{b,2}, K_{b,3})$ are employed as the positive gains in Equation (12). $\mathbf{e} = \boldsymbol{\eta}_d - \boldsymbol{\eta}$ is defined as the error of Euler angles. $\dot{\mathbf{e}} = \dot{\boldsymbol{\eta}}_d - \dot{\boldsymbol{\eta}}$ is defined as the derivative of $\mathbf{e}$. $\boldsymbol{\eta}_d$, $\dot{\boldsymbol{\eta}}_d$, and $\ddot{\boldsymbol{\eta}}_d$ are the desired Euler angle, the desired angular velocity, and the desired angular acceleration, respectively.

Substitute Equation (12) into Equation (8), we can obtain the error dynamics as

$$\ddot{\mathbf{e}} + \mathbf{K}_d \dot{\mathbf{e}} + \mathbf{K}_p \mathbf{e} + \mathbf{E}_e = 0 \tag{13}$$

where $\mathbf{E}_e = \mathbf{N} - \hat{\mathbf{N}} = (E_{e,\phi}, E_{e,\theta}, E_{e,\psi}) = (E_{e,1}, E_{e,2}, E_{e,3})$ is called the TDE error. Then, if $\bar{\mathbf{J}}$ in Equation (12) is chosen to satisfy the following condition [24,25]

$$\|\mathbf{I} - \mathbf{J}^{-1}\bar{\mathbf{J}}\|_2 < 1 \tag{14}$$

for all $t \geq 0$, the TDE error $\mathbf{E}_e$ can be represented as

$$\|\mathbf{E}_e\|_2 \leq E_e^* \tag{15}$$

for all $i = 1, 2, 3, 4$. In Equation (14), $\mathbf{I}$ is the identity matrix. $E_e^*$ is upper bound of the TDE errors, and its proof is given in [25,26]. If the sampling period is sufficiently small, the estimation in Equation (10) implies that $\hat{\mathbf{N}}$ can be as close to $\mathbf{N}$ as possible. However, as a troublesome matter of TDC scheme in real quadrotor system, $\hat{\mathbf{N}}$ cannot be estimated exactly even for small sampling period because of nonlinear disturbances, e.g., Coulomb friction, as well as to a limited sampling period from computing device. Furthermore, the undesirable side effects caused by abrupt change of the ED may be a matter of great concern to the quadrotor system because this system has a small MOI while being light-weight.

*3.2. Proposed Extended Time-Delayed Control Scheme*

To remedy the above-mentioned problems, we propose an extended time-delayed control (ETDC) scheme as a simple and powerful control scheme:

$$\boldsymbol{\tau}_\eta = -\bar{\mathbf{J}}\ddot{\boldsymbol{\eta}}_- + \bar{\boldsymbol{\tau}}_{\eta,-} + \bar{\mathbf{J}}\left[\ddot{\boldsymbol{\eta}}_d + \mathbf{K}_d \dot{\mathbf{e}} + \left(\mathbf{K}_s + \boldsymbol{\delta}_\eta |\mathbf{s}^\dagger|\right)\mathbf{s}\right] \tag{16}$$

where

$$\boldsymbol{\delta}_\eta = \mathrm{diag}\left(|E_{e,1,-}|, |E_{e,2,-}|, |E_{e,3,-}|\right) = \mathrm{diag}(\delta_\phi, \delta_\theta, \delta_\psi) \tag{17}$$

is the time-varying switching gain. Given that $\boldsymbol{\delta}_\eta$ is one-sample delayed value of $\mathbf{E}_e$, it provides the adaptation effects without additional parameters compared to the TDC scheme (Equation (12)). It implies that $\boldsymbol{\delta}_\eta$ can provide a positive effect in the proposed ETDC scheme when the sampling period is small sufficiently. $\mathbf{K}_d = \mathrm{diag}(K_{d,\phi}, K_{d,\theta}, K_{d,\psi}) = \mathrm{diag}(K_{d,1}, K_{d,2}, K_{d,3})$ and $\mathbf{K}_s = \mathrm{diag}(K_{s,\phi}, K_{s,\theta}, K_{s,\psi}) = \mathrm{diag}(K_{s,1}, K_{s,2}, K_{s,3})$ are employed to achieve the dominant pole as the positive gains. The sliding variable $\mathbf{s} = [s_\phi, s_\theta, s_\psi]^T$ is defined as

$$\mathbf{s} = \dot{\mathbf{e}} + \mathbf{K}_d \mathbf{e} \tag{18}$$

where $\mathbf{K}_d$ is the same as the pole gain applied in Equation (16). If $\mathbf{s}$ is zero, the tracking error $\mathbf{e}$ goes to zero monotonically, and its convergence rate can be adjusted by $\mathbf{K}_d$. As seen in Equation (16),

$|\mathbf{s}^{\dagger}|$ is represented as $\mathrm{diag}\left(\frac{1}{|s_{\phi}|}, \frac{1}{|s_{\theta}|}, \frac{1}{|s_{\psi}|}\right)$ equal to $\mathrm{diag}\left(\frac{1}{|s_1|}, \frac{1}{|s_2|}, \frac{1}{|s_3|}\right)$. It is defined in this paper that the following property holds $|\mathbf{s}^{\dagger}|\mathbf{s} = 1$ when $\mathbf{s} = 0$.

**Theorem 1.** *For a quadrotor system (Equation (4)) controlled by Equation (16), the sliding variables (Equation (18)) enter near the sliding manifold, i.e., $\|\mathbf{s}\|_2 \leq \varepsilon^M$, and then they are guaranteed to be uniformly ultimately bounded (UUB) as follows:*

$$\|\mathbf{s}\|_2 \leq \varepsilon^M = \max\left(\sqrt{2V_0}, \frac{\max_i(|E_i| - |E_{i,-}|)\sqrt{3}}{\min_i(K_{s,i})}\right)$$

*for all $i = 1, 2, 3$. $\varepsilon^M$ and $V_0$ are defined as the upper bound of the sliding variable and the initial value of Lyapunov function, respectively.*

**Proof.** The proof of stability is given in Appendix A.  □

**Remark 1.** *The time-varying switching gains employed in the proposed ETDC scheme makes full use of the one-sample delayed information of the TDE errors, which can suppress the TDE errors appropriately without the chattering and fluctuation. However, given that it employs the delayed information, the proposed ETDC scheme may not respond for a while in the system with low frequency. Fortunately, since the quadrotor systems have been recently developed to operate at a fast frequency, the proposed one can avoid the aforementioned problem sufficiently when implemented in the quadrotor systems. Moreover, it provides the simple structure while not requiring additional parameters compared to the TDC scheme. It implies that practical engineers can easily apply the proposed ETDC scheme to the quadrotor systems, and its results can be observed in Section 4.3.*

*3.3. Comparison with Conventional Time-Delayed Control Scheme*

$$\boldsymbol{\tau}_{\eta} = -\bar{\mathbf{J}}\ddot{\boldsymbol{\eta}}_{-} + \boldsymbol{\tau}_{\eta,-} + \bar{\mathbf{J}}\Big[\ddot{\boldsymbol{\eta}}_d + \mathbf{K}_d\dot{\mathbf{e}} + \underbrace{\left(\mathbf{K}_s + \delta_{\eta}|\mathbf{s}^{\dagger}|\right)\mathbf{s}}_{\text{Auxiliary control}}\Big] \tag{19}$$

It is noted that the proposed ETDC scheme has an effect of providing an auxiliary control. The auxiliary control scheme adjusted according to the time-varying switching gain (Equation (17)) which serves as an assistant to suppress the tracking errors even further without additional parameters. Moreover, $\mathbf{K}_s$ can be designed to be dominant in terms of the magnitude of the sliding variable. From these properties, the proposed ETDC scheme aims at improving the tracking performance with enhanced robustness while suppressing the TDE errors.

## 4. Simulation

*4.1. Simulation Setup*

To illustrate the effects of the proposed ETDC scheme, we conducted simulations through the "+"-type quadrotor system with four rotors (Equation (4)). The system model parameters are chosen to be $J_{xx} = J_{yy} = 0.08$, $J_{zz} = 0.07$, and $L = 0.010$. The parameters of the proposed ETDC scheme are set to be $\bar{J}_{\phi} = \bar{J}_{\theta} = 0.08$, $\bar{J}_{\psi} = 0.07$, $K_{s,\phi} = K_{s,\theta} = K_{s,\psi} = 4$, $K_{d,\phi} = 1$, $K_{d,\theta} = 1.5$, and $K_{d,\psi} = 3.5$. Tuning method for these parameters is introduced in Appendix B.

### 4.2. Simulation Description

The objective of these simulations is to make the roll, pitch, and yaw angles $\eta$ follow the desired trajectories $\eta_d$ that is set to zero for all angles. To demonstrate the effectiveness of the proposed ETDC scheme, we employ

- Proportional-integral-derivative (PID) control scheme
- Conventional TDC scheme [24]

in this section. It can be observed that all parameters of these control schemes are represented in Appendix C. We have tried three kinds of simulations:

(C1)　All control parameters are set to be tuned to the zero reference trajectories without the ED, as shown in Figure 2a, i.e., $\tau_d = 0$. Nominal trajectory-tracking performances of all control schemes have been demonstrated.

(C2)　The ED is applied with regard to external pressure in this simulation, which is being increased continuously as shown in Figure 2b. Then, to illustrate the effectiveness of the time-varying switching gains in the proposed ETDC scheme, the trajectory-tracking performance of the proposed one is analyzed in accordance to the sampling period, i.e., 10 ms, 30 ms, and 50 ms.

(C3)　All control parameters are also set to be tuned in the zero reference trajectories, i.e., $\tau_d = 0$. After that, to evaluate the robust trajectory-tracking performance of all control schemes, the ED is added after 6 sec, which has the non-smooth points as shown in Figure 2c. It serves to significantly disturb the motion of the quadrotor system, which has anomalous direction.

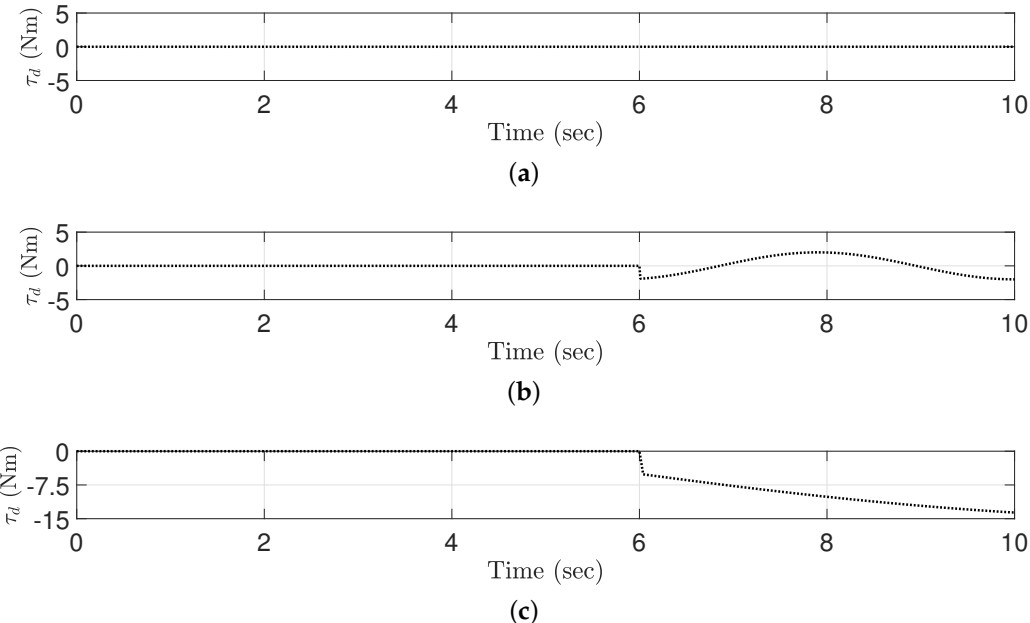

**Figure 2.** The external disturbances impacted on all axes of quadrotor system: (**a**) without ED (C1); (**b**) with ED (one direction) (C2); (**c**) with ED (anomalous direction) (C3).

### 4.3. Simulation Results

Figure 3 shows the switching gains of the proposed ETDC scheme that employ one-sample delayed information of the TDE errors. It can be observed that the magnitude of the switching gains is similar in that of the TDE errors. Therefore, the proposed ETDC scheme can appropriately respond to undesirable side effects generated by the TDE errors. In Equation (13), it provides a result in the desired error dynamics as follows:

$$\ddot{\mathbf{e}} + \mathbf{K}_d\dot{\mathbf{e}} + \mathbf{K}_p\mathbf{e} \cong 0. \tag{20}$$

This procedure offers fast convergence rate with robustness owing to dominant pole, and its results are represented in Figure 5.

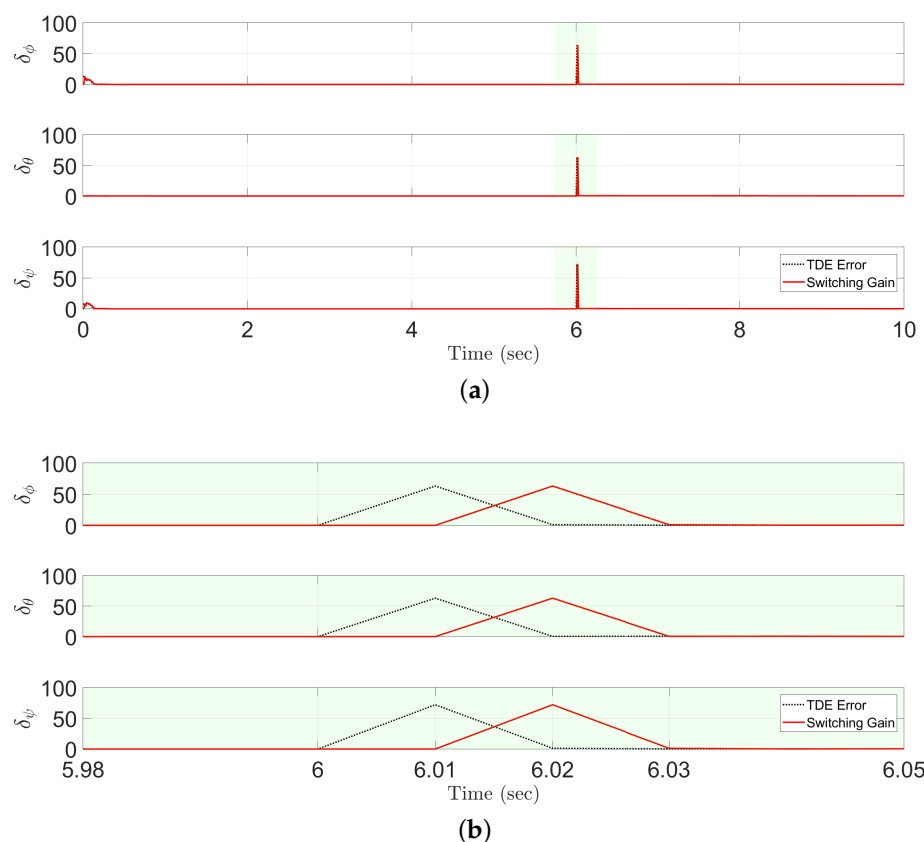

**Figure 3.** Comparison of the time-varying switching gains of the proposed ETDC scheme (solid line) and the TDE errors (dotted line): (**a**) overall scale; (**b**) partial scale.

Figure 4 shows the trajectory-tracking performance the proposed ETDC scheme according to a change in the sampling period. Given that the proposed one employs the switching gains that use one-sample delayed information of the TDE errors, it can be observed that the trajectory-tracking performance of the proposed ETDC scheme is inversely proportional to the magnitude of the sampling period. Recently, the quadrotor system is being developed to be possible in small sampling period so that such estimation in the proposed ETDC scheme will provide positive effects when implemented in a digital device.

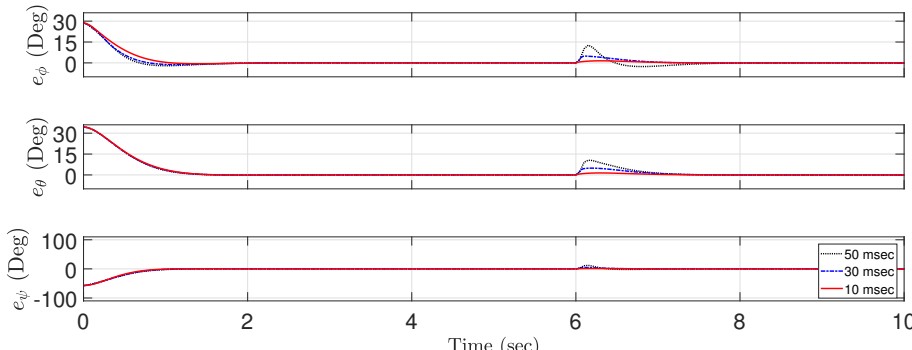

**Figure 4.** The trajectory-tracking performance of the proposed ETDC scheme in accordance with the sampling time 10 ms (solid line), 30 ms (dashed line), and 50 ms (dotted line).

Figure 5 shows trajectory-tracking errors of the PID control scheme, conventional TDC scheme, and the proposed ETDC scheme in three kinds of simulations:

(C1)  Figure 5a shows a result in zero reference trajectory without the ED. As seen in Figure 2a, all control schemes have the similar level in trajectory-tracking performance. It implies that they have no significant difference in performance of reference trajectory without the ED. The root mean square (RMS) values of the trajectory-tracking errors are given in Table 1.

(C2)  Figure 5b shows a result in nominal trajectory-tracking performance while generating the ED. To illustrate the undesirable side effects generated by the abrupt ED (Figure 2b), a sinusoidal signal is added in the simulation procedure. The signal have strong external pressure instantaneously. As seen in Figure 5b, it can be observed that both the conventional TDC scheme and the proposed ETDC scheme work better than the PID control scheme in case of increasing ED continuously after 6 s. In detail, the proposed ETDC scheme has better performance than the conventional TDC scheme in vicinity of 6 s. It means that the proposed one provides precise trajectory-tracking performance while enhancing the robustness. The RMS values of the trajectory-tracking errors are given in Table 2.

(C3)  Figure 5c shows the trajectory-tracking errors in the external pressure with high frequency trajectory, i.e., a sinusoidal signal $2\sin(1.5t)$, unlike in Figure 5b. The ED causes negative results significantly in both the PID control scheme and the conventional TDC scheme. On the other hand, Figure 5c represents that the proposed ETDC scheme has improved the robustness compared to other control schemes. The RMS values of the trajectory-tracking errors are given in Table 3.

**Table 1.** RMS values of trajectory-tracking errors (Figure 5a) measured from 5 s to 10 s.

| Control Strategies | Roll, $\phi$ (Deg) | Pitch, $\theta$ (Deg) | Yaw, $\psi$ (Deg) |
|---|---|---|---|
| PID control scheme | $2.20 \times 10^{-5}$ | $1.94 \times 10^{-5}$ | $2.32 \times 10^{-8}$ |
| Conventional TDC scheme [24] | $1.61 \times 10^{-5}$ | $0.82 \times 10^{-5}$ | $2.29 \times 10^{-8}$ |
| Proposed ETDC scheme | $1.40 \times 10^{-5}$ | $0.51 \times 10^{-5}$ | $1.69 \times 10^{-8}$ |

**Table 2.** RMS values of trajectory-tracking errors (Figure 5b) measured from 5 s to 10 s.

| Control Strategies | Roll, $\phi$ (Deg) | Pitch, $\theta$ (Deg) | Yaw, $\psi$ (Deg) |
|---|---|---|---|
| PID control scheme | 46.03 | 47.70 | 25.07 |
| Conventional TDC scheme [24] | 2.46 | 2.39 | 1.54 |
| Proposed ETDC scheme | 0.37 | 0.36 | 0.25 |

**Table 3.** RMS values of trajectory-tracking errors (Figure 5c) measured from 5 s to 10 s.

| Control Strategies | Roll, $\phi$ (Deg) | Pitch, $\theta$ (Deg) | Yaw, $\psi$ (Deg) |
|---|---|---|---|
| PID control scheme | 5.89 | 5.84 | 3.37 |
| Conventional TDC scheme [24] | 1.67 | 1.61 | 1.00 |
| Proposed ETDC scheme | 1.34 | 1.25 | 0.83 |

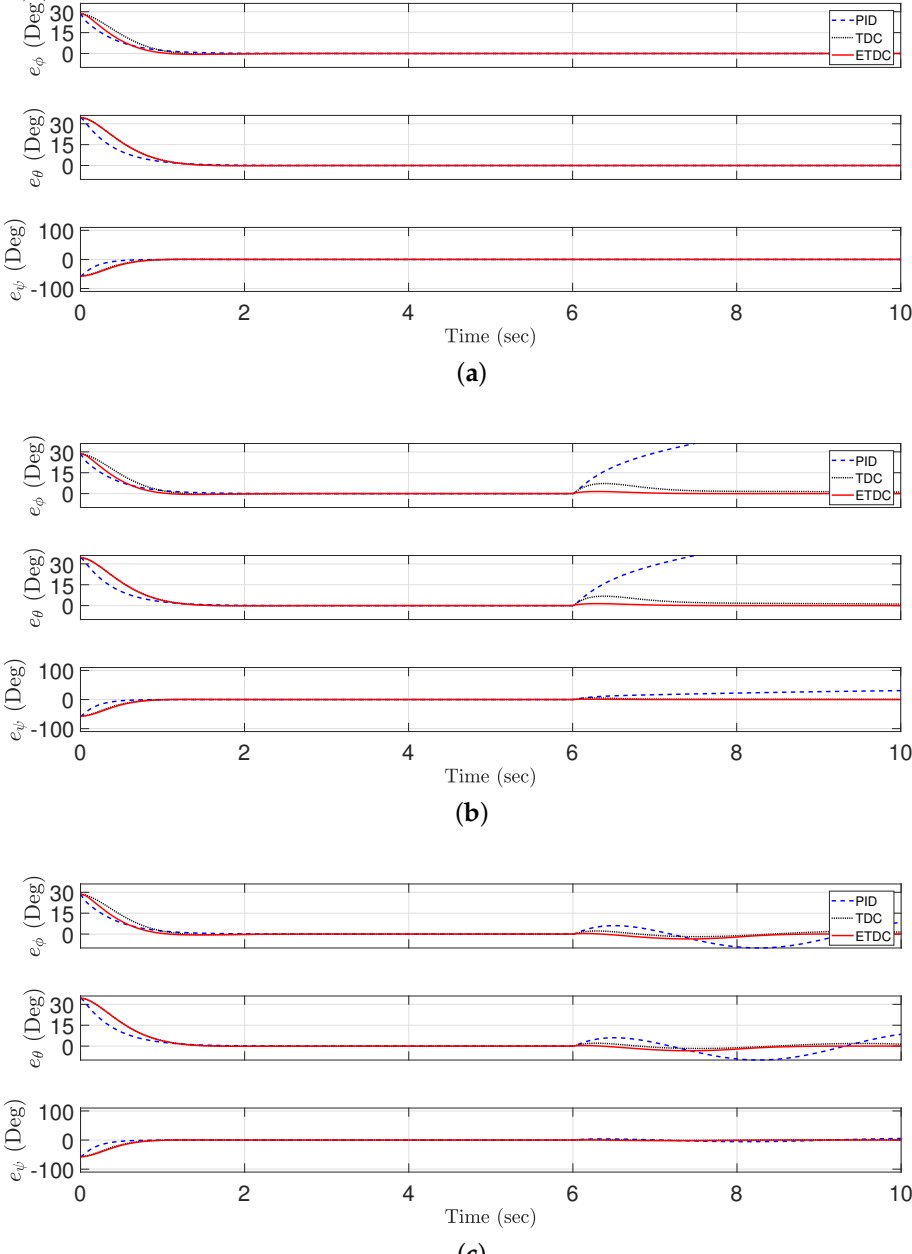

**Figure 5.** Comparison of the trajectory-tracking errors of PID control scheme (dotted line), conventional TDC scheme (dashed line), and the proposed ETDC scheme (solid line): (**a**) without ED (C1); (**b**) with ED (one direction) (C2); (**c**) with ED (anomalous direction) (C3).

## 5. Discussion

### 5.1. Future Perspective

In [27–29], the role of biological findings can be used to optimize current control model systems. As future perspective, it would be meaningful to employ the role of biological findings for wide readership of several papers, including this paper.

### 5.2. Supplementary Simulation

In this paper, no experimental research is conducted. However, in order to investigate more realistic conditions, see, for example [30], we employ the quadrotor system assuming measurement uncertainties.

Parameters of all control schemes are set to be tuned as in the scenario (C1) of Section 4. We use the simulation with the quadrotor system while adding the internal disturbance (ID), e.g., random noise, in the scenario (C2) of Section 4. Then, we evaluate the trajectory-tracking performance of all control schemes.

Figure 6 shows trajectory-tracking errors of the PID control scheme, conventional TDC scheme, and the proposed ETDC scheme, which describes the same as criteria for the scenario (C2) in Section 4. Since this simulation have an undesirable side effect on the random noise, i.e., mean = 0.05 and variance = 0.05, all control schemes represents the degraded trajectory-tracking performance in some areas when compared with Figure 5b. At first, as seen in Figure 6, both the proposed ETDC scheme and conventional TDC scheme are not instantaneously convergent in the vicinity of 1 s, i.e., green circle. In other words, it implies that they cause an overshoot signal because of their mathematical structure. However, the proposed ETDC scheme aims at providing precise trajectory-tracking performance against the noise due to the proposed time-varying switching gain. Next, it can be observed that the PID control scheme and conventional TDC scheme are not guaranteed to be hovering in the vicinity of 10 s, i.e., magenta circle. In particular, the PID control scheme causes fluctuation in the vicinity of 10 s because of the unattractive effect generated by the ED and the ID. As a result, it may lead to unstable motion in the quadrotor system. On the other hand, the proposed ETDC scheme is working well for improving convergence rate while guaranteeing the system stability, unlike these existing control schemes, as seen in Figure 6.

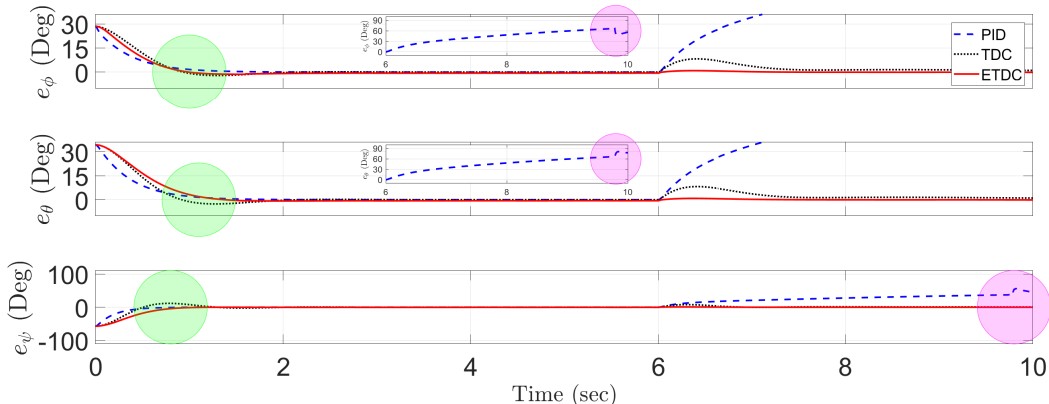

**Figure 6.** Comparison of the trajectory-tracking errors of PID control scheme (dotted line), conventional TDC scheme (dashed line), and the proposed ETDC scheme (solid line): with ED (one direction) (C2) and ID.

## 6. Conclusions

The proposed ETDC scheme was proposed to improve both nominal and robust trajectory-tracking performance in the quadrotor system. The proposed one does not require any number of additional tuning parameters when compared to a well-known conventional TDC scheme, and hence we can still keep it simple. Moreover, since the TDE errors can be suppressed appropriately by the proposed time-varying switching gains, the proposed ETDC scheme provides a fast convergence rate while guaranteeing the system stability, even in the case of various external disturbances. As a result, the proposed ETDC scheme aims at providing precise trajectory-tracking performance with robustness against the abrupt ED unlike the TDC scheme. It is shown through the simulations, and then the trajectory-tracking errors are guaranteed to be UUB while not having undesirable side effects.

The proposed ETDC scheme could be a good replacement of the conventional TDC scheme. We believe that it would be a good trial to increase ease of use and improve the trajectory-tracking performance.

**Author Contributions:** Conceptualization: J.B.; software: J.B.; data curation: J.B.; writing—original draft preparation: J.B.; writing—review and editing: J.B.; supervision: J.J.; project administration: J.J.; funding acquisition: J.J. All authors have read and agreed to the published version of the manuscript.

**Funding:** This work was supported by the University of Suwon, 2017.

**Conflicts of Interest:** The authors declare no conflict of interest.

## Appendix A. Proof of Stability

To facilitate the proof in Equation (16), the Lyapunov function, denoted by $V$, is defined as follows:

$$V = \frac{1}{2}\mathbf{s}^T\mathbf{s}. \tag{A1}$$

Then, the time derivative of Equation (A1) can be obtained as

$$
\begin{aligned}
\dot{V} &= \mathbf{s}^T\dot{\mathbf{s}} \\
&= \mathbf{s}^T[\ddot{\mathbf{e}} + \mathbf{K}_d\dot{\mathbf{e}}] \\
&= \mathbf{s}^T[\ddot{\boldsymbol{\eta}}_d - \ddot{\boldsymbol{\eta}}_- + \mathbf{K}_d\dot{\mathbf{e}}]
\end{aligned} \tag{A2}
$$

where $\dot{\mathbf{e}} = \dot{\boldsymbol{\eta}}_d - \dot{\boldsymbol{\eta}}$ is the derivative of Euler angles, and its derivative is $\ddot{\mathbf{e}} = \ddot{\boldsymbol{\eta}}_d - \ddot{\boldsymbol{\eta}}$. Substituting Equation (10) into Equation (A2), we have

$$\dot{V} = \mathbf{s}^T\left(\ddot{\boldsymbol{\eta}}_d - \mathbf{N} + \bar{\mathbf{J}}^{-1}\boldsymbol{\tau}_{\eta,t} + \mathbf{K}_d\dot{\mathbf{e}}\right). \tag{A3}$$

Substituting $\boldsymbol{\tau}_{\eta,t}$ in Equation (A3) yields

$$
\begin{aligned}
\dot{V} &= \mathbf{s}^T\left(-\mathbf{N} + \mathbf{N}_- - \mathbf{K}_s\mathbf{s} - \delta_\eta|\mathbf{s}^\dagger|\mathbf{s}\right) \\
&= \mathbf{s}^T\left(-\mathbf{E}_e - \mathbf{K}_s\mathbf{s} - \delta_\eta|\mathbf{s}^\dagger|\mathbf{s}\right) \\
&\leq \sum_{i=1}^{3}|s_i||E_{e,i}| - \sum_{i=1}^{3}K_{s,i}s_i^2 - \sum_{i=1}^{3}|s_i||E_{e,i,-}|
\end{aligned} \tag{A4}
$$

where $\mathbf{E}_e$ is upper-bounded according to Equation (14). It follows then that

$$
\begin{aligned}
\dot{V} &\leq -\sum_{i=1}^{3}K_{s,i}s_i^2 + \sum_{i=1}^{3}|s_i|(|E_{e,i}| - |E_{e,i,-}|) \\
&\leq -\min_i(K_{s,i})\sum_{i=1}^{3}s_i^2 + \max_i(|E_{e,i}| - |E_{e,i,-}|)\sum_{i=1}^{3}|s_i|.
\end{aligned} \tag{A5}
$$

From the second term of the right-hand side of Equation (A5), it follows that we have

$$\dot{V} \le -\min_i(K_{s,i}) \sum_{i=1}^{3} s_i^2 + \max_i(|E_i| - |E_{i,-}|)\sqrt{6V}. \tag{A6}$$

From Barbalat's lemma [31], we can represent $\|\mathbf{s}\|_2 \to 0$ as $t \to \infty$ when $\max_i(|E_i| - |E_{i,-}|)$ is zero in Equation (A6). As the worst case of Equation (A6), we assume that the right-hand side is larger than or equal to zero. It follows that

$$0 \le -\min_i(K_{s,i}) \sum_{i=1}^{3} s_i^2 + \max_i(|E_i| - |E_{i,-}|)\sqrt{6V}. \tag{A7}$$

From Equation (A7), the Lyapunov function $V$ is upper-bounded as

$$V \le \max\left(V_0, \left(\max_i(|E_i| - |E_{i,-}|)\right)^2 \cdot \frac{3}{2\left(\min_i(K_{s,i})\right)^2}\right) \tag{A8}$$

which means that

$$\|\mathbf{s}\|_2 \le \max\left(\sqrt{2V_0}, \frac{\max_i(|E_i| - |E_{i,-}|)\sqrt{3}}{\min_i(K_{s,i})}\right) \tag{A9}$$

where $V_0$ is initial value of Lyapunov function. The upper bound in Equation (A9) is directly dependent on $\mathbf{K}_s$. In other words, it can be observed that the larger $\mathbf{K}_s$, the smaller $\|\mathbf{s}\|_2$. From Equation (A7), $\|\mathbf{s}\|_2$ is UUB so that the tracking error $\mathbf{e}$ is also UUB owing to bounded-input bounded-output stability [31] from Equation (18).

**Appendix B. Parameters Tuning of Proposed Extended Time-Delayed Control Scheme**

(S1)　To begin with, $\mathbf{K}_d$ and $\mathbf{K}_s$ in Equation (16) should be chosen to provide desirable error dynamics by the pole assignment when the TDE errors are assumed to be zero, i.e., $\delta_\eta = 0$. Then, $\mathbf{K}_d$, and $\mathbf{K}_s$ are chosen to obtain dominant pole, and their initial values are specified as identity matrix $\mathbf{I}$.

(S2)　Starting off from initial value $\bar{\mathbf{J}} = 0.0001$ in Equation (16), tuning it may be tractable because the inertial moment of a quadrotor system hardly changes. However, if the $\bar{\mathbf{J}}$ is too large, the trajectory-tracking performance will be degraded due to the noise effect generated by angular acceleration in Equation (16).

(S3)　After a standard setup, please check the imaginary motion of $\mathbf{s}$ in Equation (18) from the plot. Then, $\mathbf{K}_d$ should be tuned to adjust the convergence rate of the $\mathbf{s}$. It implies that the $\mathbf{K}_d$ may be increased for fast convergence rate, and hence the poles may be slightly shifted from the imaginary axis.

(S4)　In order to guarantee the dominant pole, please align $\mathbf{K}_s$ with $\mathbf{K}_d$.

(S5)　Please, repeat (S3) $\sim$ (S4) once again for achieving the desired level.

**Appendix C. Parameters of All Control Schemes in Simulation**

All parameters of control schemes introduced in the Section 4 can be represented as follows:

(1)　PID control scheme

　　– P-gain: $K_{pp,\phi} = K_{pp,\theta} = 10$, $K_{pp,\psi} = 20$
　　– I-gain: $K_{pi,\phi} = K_{pi,\theta} = K_{pi,\psi} = 5$
　　– D-gain: $K_{pd,\phi} = K_{pd,\theta} = K_{pd,\psi} = 4$.

(2)　Conventional TDC scheme

- $\bar{J}_\phi = \bar{J}_\theta = 0.08$, $\bar{J}_\psi = 0.07$
- $K_{a,\phi} = K_{a,\theta} = K_{a,\psi} = 8$
- $K_{b,\phi} = K_{b,\theta} = K_{b,\psi} = 16$.

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
