# Peer review of "A Model-Free Control Scheme for Attitude Stabilization of Quadrotor Systems"

_electronics, doi:10.3390/electronics9101586_

Round 1
Reviewer 1 Report
1. In my opinion the current version of the paper is improved. However, I cannot find a discussion how to estimate the disturbance \hat{N}, Eq (10). I think that it is important to assume that acceleration \ddot\eta is available for measurement.
2. Eq. (1) is incorrect (this mistake can often be found in the literature). The gradients \partial L/\partial q, \partial L/\partial \dot{q} are row vectors, while \tau_\eta i a column vector. As a result, the given derivatives have to be transposed.
3. Typos/errors:
- The sentence below Eq. (12) is unfinished. „Then, if … for all t\geq 0.” ???
- line 199: In this paper, it is hard to obtain realistic equipments unlike [29]...” I think it would be better to state that:
“In this paper no experimental research is conducted. However, in order to investigate more realistic conditions, see for example [29], we employ the quadrator system assuming measurement uncertainties”.
- line 125:
„Moreover, it provides the the simple” - double „the”
Reviewer 2 Report
The paper can be accepted for publication in present form.
Author Response
Please see the attachment.

This manuscript is a resubmission of an earlier submission. The following is a list of the peer review reports and author responses from that submission.
Round 1
Reviewer 1 Report
This paper deals with the problem of practical model-free control scheme for attitude stabilization of unmanned aerial vehicle systems, and the effectiveness of the method is verified by simulation results. However, there are some problems in the paper as follows.
(1) There are many related results in literatures, even in Electronics, and the authors should give a detailed overview of the state of the art in quadrotor control in Section 1. In addition, the novelty and the contribution of the paper should be further clarified.
(2) In Section 2, the description of the model can be simplified by citing proper references.
(3) Section 3 can be separated into two parts, i.e., TDC scheme and ETDC scheme.
(4) In Section 3, the improvement of ETDC scheme should be discussed compared with TDC scheme. And also, parameter tuning guidelines should be given for ETDC scheme.
(5) In Equation (16), the authors should discuss how to realize |s-| when s=0.
(6) Only simulation results are not convincing for the verification of the proposed method, and experimental results are strongly suggested. Even if experimental results are not available, the authors should consider much more practical factors in simulation, such as measurement noises, internal and external disturbances, etc.
(7) Parameters for PID and TDC also should be given in the simulation. And also, the control gains of three controllers should be chosen for fair comparison.
(8) The writing of the paper needs to be polished.
Reviewer 2 Report
The article “A Practical Model-free Control Scheme for Attitude Stabilization of Unmanned Aerial Vehicle Systems” by Baek and Jung describes the extended time-delayed control (ETDC) scheme, a model-free control scheme used to address trajectory-tracking control and that can be applied to unmanned aerial vehicles.
Although the article is rather interesting, I cannot understand the novelty of the model proposed by authors and their original contribution to the field.
Also, can be interesting for authors, to mention in a “future perspective” section in their discussion, the role of biological findings that can be used to optimize current control model systems. This would greatly improve the appeal for the wide readership of this Journal.
Few examples that I strongly encourage to comment and include in the work are:
Encoding lateralization of jump kinematics and eye use in a locust via bio-robotic artifacts. Journal of Experimental Biology, 222(2), (2019). Doi: 10.1242/jeb.187427
Development and experiments of a bio-inspired robot with multi-mode in aerial and terrestrial locomotion. Bioinspiration & biomimetics, 14(5), 056009 (2019).
A novel autonomous, bioinspired swimming robot developed by neuroscientists and bioengineers. Bioinspiration & biomimetics, 7(2), 025001, (2012).
A careful English revision is need throughout the text.
I hope my suggestions can help authors to improve their manuscript.
Author Response
Please see the attachment (Pages 5 and 6).

Reviewer 3 Report
- It is unclear why the considered method is dedicated for an UAV system. In fact, this approach can be used for other fully actuated mechanical systems. In my opinion, the paper deals with the application of some kind of partially model-free control approach supported by sliding modes. Conversely, only some kind of UAVs, such as quad-rotors, can be controlled in this way. For example, I think that the considered controller cannot be directly employed for autonomous helicopters and airplanes.
- Although at places practical aspects are mentioned, authors do not provide an efficient way to estimate TDE error. From Eq. (11) it follows that the acceleration \ddot\eta_ has to be measured. However, I am afraid that such a measurement can be difficult in practice due to the presence of noise.
- In simulations only a rapid increase of gains \delta_\eta is visible when non-smooth transition of \tau_d occurs. Moreover, one can state that in simulations TDE error can be found easily. Unfortunately, no experimental results are presented. As a result, it is difficult to support the statement that the algorithm is effective in practice. Thus, I suggest to consider simulations where a measurement noise is taken into account. It would help to demonstrate the controller performance in more realistic scenarios. In addition, control inputs should be also presented.
- Notation of s^- is unclear and its definition in lines 114-115 is confusing. In particular, how to interpret „-” sign? Does it indicate the delayed value of s (it can be found accoriding to the stability proof presented in the Appendix)?
- I think that the stability proof could be modified in order to consider the convergence of s(t). Currently, inequality (A8) is unclear since it does not indicate how s(t) converges to a vicinity of zero. Please notice that even when E_{i}-E_{i,-} = 0 from (A8) and (A9) one cannot conclude that s tends to zero.
- Other comments:
- Section 2 can be shorten as it contains a well-known model of typical quadrotor.
- Eq. (1) – the size of brackets should be increased.
- Eq. (10) – angles \phi, \theta and \psi should be written in normal (not bold) font.
- Eq. (A8) – E_{i-L} should be replaced by E_{i,-}.
Author Response
Please see the attachment (Pages 7-9).

Round 2
Reviewer 1 Report
The paper has been modified according to the reviewers' comments in last round, and can be accepted for publication in present form.
Reviewer 2 Report
Authors addressed most of my suggestions.
Reviewer 3 Report
I think that the paper has been improved correctly. However, it could be interesting to consider performance of the algorithm assuming more realistic measurement conditions. Since there are no experimental results, it is hard to say what are properties of this control method in practice.
Authors might also want to look at recent publication:
https://ieeexplore.ieee.org/abstract/document/9093153
where experimental validation of a robust algorithm for trajectory tracking is considered.